# Changes in tobacco sales before, during, and after the COVID-19 pandemic in Japan: An interrupted time series analysis

Kanae Kondo[1,2*], Ichizo Morita[2], Shigemitsu Sakuma[3], Isao Ohsawa[1]

1 Faculty of Health Science, Aichi Gakuin University, Nisshin, Aichi, Japan, 2 Japanese Red Cross Toyota College of Nursing, Toyota, Aichi, Japan, 3 Department of Fixed Prosthodontics and Oral Implantology, School of Dentistry, Aichi Gakuin University, Nagoya, Aichi, Japan

* k-kondo@rctoyota.ac.jp

## Abstract

### Objectives

This study aimed to longitudinally examine nationwide changes in smoking behavior among the Japanese population in response to the COVID-19 pandemic.

### Design and setting

A secondary analysis was conducted using cigarette sales data from the Tobacco Institute of Japan, monthly tobacco expenditures from the Family Income and Expenditure Survey conducted by the Ministry of Internal Affairs and Communications, and smoking prevalences from the National Database of Open Data Japan. An interrupted time-series segmented regression model adjusted for seasonality and autocorrelation was used to examine changes in cigarette sales and monthly tobacco expenditures before and after the first declaration of a state of emergency.

### Participants

Depending on the data source, data from 2014 to 2022, from 2015 to 2025, or from 2015 to February 2026 were used.

### Primary and secondary outcome measures

Changes in level and slope were evaluated before and after the first state of emergency and after COVID-19 was downgraded to a common infectious disease.

### Results

Cigarette sales declined at a significantly slower rate after the first state of emergency than before. Price-adjusted monthly tobacco expenditures increased significantly by 132 JPY compared with the previous month, although the expenditure findings were less robust in the quadratic sensitivity analysis. The smoking prevalence among

**Data availability statement:** The dataset used to support the findings of this study is publicly available in Zenodo: https://doi.org/10.5281/zenodo.20324787. This study used publicly available aggregate data from third-party sources, including quarterly cigarette sales data from the Tobacco Institute of Japan, monthly household tobacco expenditure data and cigarette price data from the Ministry of Internal Affairs and Communications, and annual smoking prevalence data from the NDB Open Data provided by the Ministry of Health, Labour and Welfare. No individual-level data were used in this study.

**Funding:** The author(s) received no specific funding for this work.

**Competing interests:** The authors have declared that no competing interests exist.

men declined slightly each year, from 34.2% in 2014 to 31.9% in 2022. In contrast, the smoking prevalence among women remained relatively stable, ranging between 9.4% and 9.9%.

## Conclusions

Although cigarette sales declined more slowly and household tobacco expenditures increased during the first state of emergency in Japan, smoking prevalences did not change substantially. These findings suggest changes in tobacco purchasing patterns, including stockpiling, and possible changes in smoking intensity among continuing smokers, rather than a meaningful population-level decline in smoking behavior. There was no robust evidence of a clear reversal after May 2023.

## Introduction

Smoking is well known to cause many diseases and health problems [1]. Therefore, various organizations around the world have made efforts to reduce both the number of smokers and the amount they smoke [2]. These efforts range from encouraging individuals through education [3] and information provision to broader public health initiatives such as restricting smoking areas [4] and enacting anti-smoking legislation laws to curb smoking [5]. In Japan as well, smoking cessation has been promoted through the enactment of the Health Promotion Act [6]. However, while regulations are imposed on facility managers, the actual decision to refrain from smoking remains largely at the discretion of individuals.

Smoking increases the risk of malignant tumors, including lung cancer—one of the most serious health consequences associated with smoking. These risks are cumulative, and smoking rarely causes cancer immediately, with onset typically occurring 10–20 years after exposure begins [7]. Therefore, people may not perceive the health risks of smoking as immediate or important. From the early stages of the COVID-19 pandemic, it was pointed out that a history of smoking increases the risk of severe illness [8,9]. This information was disseminated to the Japanese population via the Internet and mass media.

The risk of developing life-threatening illness from COVID-19 due to smoking typically manifests within a few days to one or two weeks, in contrast to the decades-long latency period associated with smoking-related cancers. This shorter time frame for serious outcomes might have been expected to influence smokers' behavior more strongly. However, many studies have reported an increase in smoking during the COVID-19 outbreak and lockdowns [10–12]. Moreover, an analysis of national data from the United States showed a decline in smoking cessation rates in 2020 compared to previous years [13]. This presents a paradox if individuals fully understood the risks associated with smoking.

Some studies have also reported a decrease in smoking following the COVID-19 pandemic [14,15]. However, only a few have evaluated the impact of the pandemic on smoking behavior using a longitudinal design [16]. Japan's first state

of emergency did not involve a full lockdown, such as a citywide blockade or mandatory curfew, but instead included requests to limit outdoor activities and reduce interpersonal contact where feasible. Such pandemic-related changes in daily routines and coping behaviors may have influenced smoking-related behaviors during the restriction period. Understanding how the substantial societal changes associated with the COVID-19 pandemic influenced smoking behavior may provide valuable insight into effective strategies for reducing smoking. This study aimed to longitudinally clarify nationwide changes in smoking behavior among the Japanese population in response to the COVID-19 pandemic.

## Materials and methods

### Data sources

The number of cigarette sales was obtained from tobacco statistics published on the website of the Tobacco Institute of Japan [17]. These data report the actual number of cigarettes sold and the total sales amount for each quarter across Japan. In this study, the quarterly number of cigarettes sold was used for the analysis of cigarette sales trends. The study utilized data from the first quarter of 2015 to the fourth quarter of 2025.

National tobacco expenditures were obtained from the Family Income and Expenditure Survey conducted by the Ministry of Internal Affairs and Communications [18]. Monthly tobacco expenditures by households consisting of two or more members from January 2015 to February 2026 were included in the analysis. The retail price of tobacco is uniform across Japan, with price changes primarily driven by increases in the base price, tobacco tax, and consumption tax rate. Since 2015, tobacco prices have increased due to base price hikes in April 2016 and April 2017; tobacco tax increases in October 2018, October 2020, and October 2021; and a revision of the consumption tax rate in 2019. The analysis used tobacco expenditure amounts adjusted for price fluctuations due to consumption tax and tobacco tax based on cigarette prices from the Ministry of Internal Affairs and Communications' Retail Price Statistics Survey [19]. Thus, the monthly analysis was based on price-adjusted tobacco expenditures for households with two or more members.

Information on smoking prevalence from 2014 to 2022 was obtained from the National Database (NDB) of Open Data [20]. In this survey, smokers are defined as individuals who have smoked more than 100 cigarettes in total or have smoked for more than six months and reported smoking within the past month. The information on smoking status included in the NDB Open Data originates from questionnaire surveys conducted as part of government-led health check-ups, with 25–30 million individuals aged 40 years or older participating annually. Because these data were available as annual national percentages, smoking prevalence in men and women was summarized descriptively rather than analyzed using the interrupted time-series model.

This study was a secondary analysis of these data and was approved by the Ethics Committee of the Japanese Red Cross Toyota College of Nursing (Project No. 2211). The data were accessed for research purposes on from 01 August 2023–05 May 2026. The authors had no access to information that could identify individual participants during or after data collection.

### Statistical methods

An interrupted time-series (ITS) analysis [21] was conducted to examine changes in cigarette sales using a single segmented regression model with two intervention points: the first declaration of a state of emergency in mid-April 2020 (first intervention) and the reclassification of COVID-19 to a common infectious disease on May 8, 2023 (second intervention). For quarterly cigarette sales, the second intervention was aligned to the April–June 2023 quarter. The data period for analysis ranged from the first quarter (January – March) of 2015 to the fourth (October – December) quarter of 2025.

Monthly tobacco expenditures were also analyzed using ITS analysis, with April 2020 and May 2023 defined as the first and second intervention points, respectively, within the same segmented regression framework. For interpretability and to facilitate comparisons between the tables and figures, the time variable was coded as a consecutive sequence and

centered at 0 for the last pre-intervention time point (January–March 2020 for quarterly cigarette sales; March 2020 for monthly tobacco expenditures); therefore, the intercept represents the estimated outcome at these reference points. For quarterly cigarette sales, seasonality was modeled using quarter indicators (Q2–Q4), with Q1 as the reference category. For monthly tobacco expenditures, seasonality was modeled using month indicators (February–December), with January as the reference category. To account for autocorrelation, the models were fitted using linear mixed models with an AR(1) covariance structure. As a residual diagnostic, the Durbin–Watson statistic was examined in the corresponding ordinary least-squares segmented regression models. Following previous methodological guidance on segmented regression design specification [22], the level-change parameter for each intervention was defined as the immediate deviation at the first post-intervention time point from the value expected under continuation of the trend in the immediately preceding period, whereas the slope-change parameter was defined as the change in trend after the intervention relative to that immediately preceding period. In addition, as a sensitivity analysis, a quadratic time term (time²) was added to the fixed-effects part of the model to assess the robustness of the findings to possible nonlinearity in the underlying secular trend.

Smoking prevalences (men and women) from NDB Open Data (2014–2022) were summarized descriptively by year and presented as percentages; no hypothesis testing was performed.

All analyses were performed using IBM SPSS Statistics version 29.0 (IBM Corp., Armonk, NY, USA).

## Results

### Trends in cigarette sales and impact of the first state of emergency declaration and downgrade

From the first quarter of 2015 to the first quarter of 2020, cigarette sales exhibited a downward trend with periodic fluctuations (Fig 1, Table 1), decreasing by approximately 1 billion cigarettes per quarter. In the quarter preceding the first state of emergency declaration (January–March 2020), cigarette sales amounted to 26.48 billion units. After adjustment

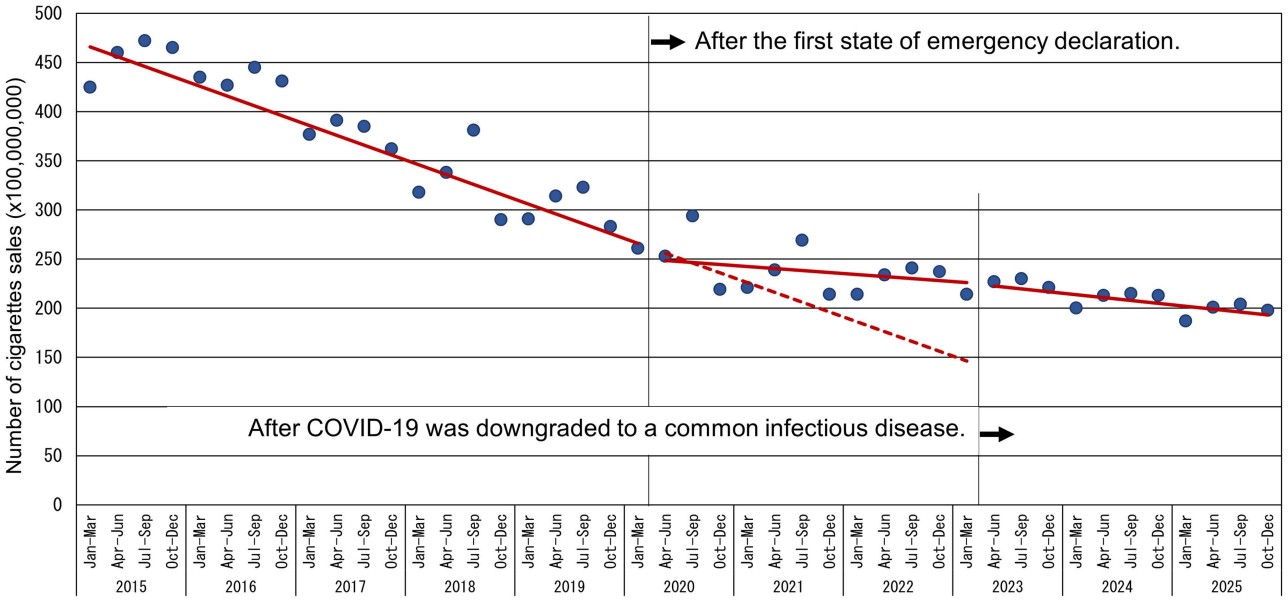

**Fig 1. Trend in cigarette sales before and after the first state of emergency declaration.** Note: The solid line shows fitted values from the interrupted time-series segmented regression model adjusted for seasonality and autocorrelation. The dashed line shows the predicted number of cigarette sales had the COVID-19 pandemic and the first state of emergency not occurred, assuming continuation of the pre-intervention trend (counterfactual). The time variable was centered at 0 for January–March 2020; thus, the intercept represents the estimated cigarette sales in that quarter.

**Table 1. Parameter estimates from the interrupted time-series segmented regression model for quarterly cigarette sales adjusted for seasonality and autocorrelation.**

| | | Estimate | 95% confidence interval | | P-value |
|---|---|---|---|---|---|
| | | | Lower limit | Upper limit | |
| Intercept | | 244.982 | 228.278 | 261.685 | <0.001 |
| Baseline trend | | −9.991 | −11.214 | −8.767 | <0.001 |
| After the first state of emergency declaration | Level change | −17.468 | −41.080 | 6.143 | 0.133 |
| | Slope change | 7.962 | 4.858 | 11.067 | <0.001 |
| After COVID-19 was downgraded to a common infectious disease | Level change | −1.481 | −29.119 | 26.157 | 0.910 |
| | Slope change | −0.924 | −5.227 | 3.379 | 0.643 |
| Q2 | | 22.254 | 8.958 | 35.551 | 0.002 |
| Q3 | | 42.881 | 29.759 | 56.002 | <0.001 |
| Q4 | | 19.110 | 5.973 | 32.248 | 0.006 |

Note: Seasonality was modeled using quarter indicators (Q2–Q4), with Q1 as the reference category. Autocorrelation was accounted for using a linear mixed model with an AR(1) covariance structure. P-values were obtained from t-tests for fixed-effect parameters in the linear mixed model, using the denominator degrees-of-freedom method specified in SPSS. Values <0.05 were considered statistically significant.

for seasonality and autocorrelation, the level change at the first intervention was not statistically significant, whereas the slope became significantly less negative after the first declaration (p < 0.001), indicating a deceleration in the declining trend. In the first quarter of 2020, cigarette sales showed little change compared to the previous quarter; however, the post-intervention trend corresponded to a smaller quarterly decline (approximately 200 million cigarettes) compared with the pre-intervention period. After the second intervention (the reclassification of COVID-19 as a common infectious disease, reflected in the April–June 2023 quarter), there was no significant change in either the level or slope of cigarette sales.

The quarter indicators were statistically significant, supporting the presence of seasonal variation in cigarette sales. The Durbin–Watson statistic in the corresponding ordinary least-squares segmented regression model was 2.100.

As a sensitivity analysis, we additionally fitted a model including a quadratic time term (time²). For quarterly cigarette sales, the quadratic term was not statistically significant, and the direction and statistical significance of the intervention-related estimates were unchanged.

## Trends in monthly tobacco expenditures

From January 2015 to March 2020, price-adjusted monthly tobacco expenditures exhibited a declining trend with periodic fluctuations, decreasing from approximately 1,000 JPY to 800 JPY (Fig 2, Table 2). After adjustment for seasonality and autocorrelation, in April 2020, when the government declared the first state of emergency, price-adjusted monthly tobacco expenditures increased significantly by 132 JPY compared to the previous month (p = 0.005). However, from that point through April 2023, the expenditures continued to decline; the change in slope after April 2020 was not statistically significant (p = 0.486), indicating no evidence that the rate of decline differed from the pre-pandemic trend. After May 2023, when COVID-19 was downgraded to a common infectious disease, the rate of decline in price-adjusted monthly tobacco expenditures became significantly less steep (p = 0.027), whereas the level change was not statistically significant (p = 0.731).

Several month indicators were also statistically significant, supporting the presence of seasonal variation in tobacco expenditures. The Durbin–Watson statistic in the corresponding ordinary least-squares segmented regression model was 1.669.

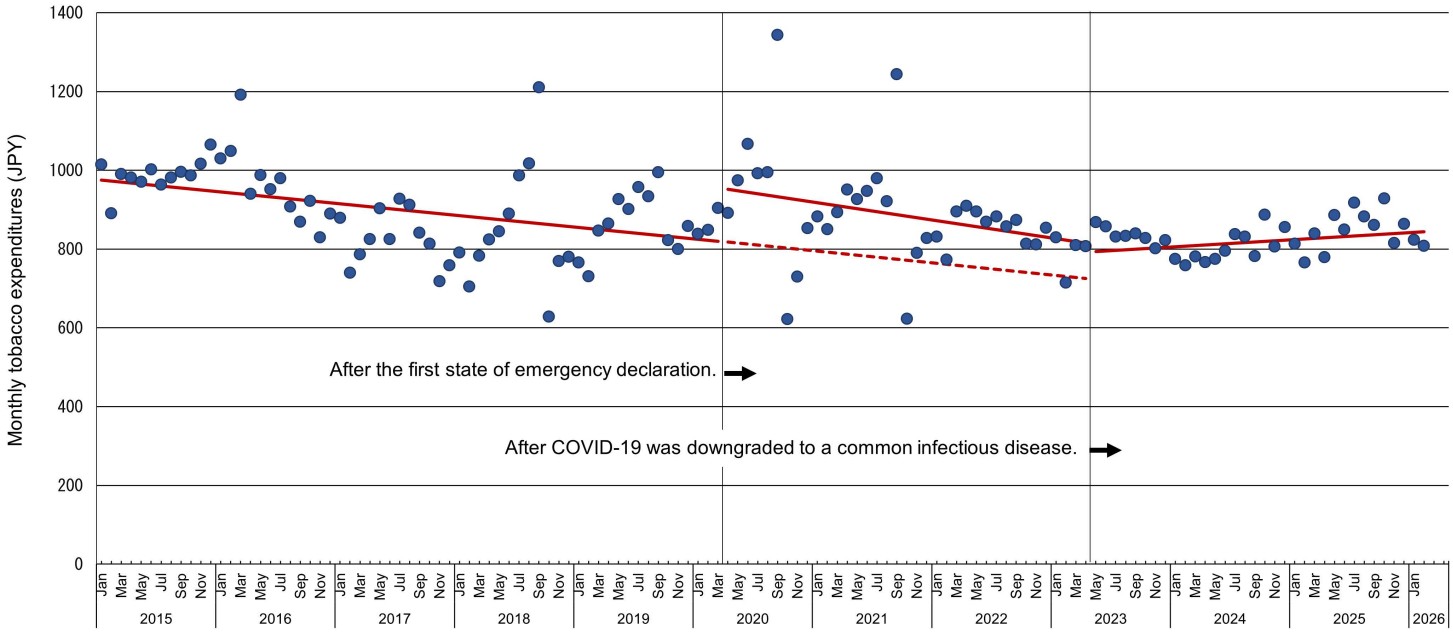

**Fig 2. Trend in monthly tobacco expenditures per household (households with two or more members) before and after the first state of emergency declaration.** Note: The round markers indicate tobacco expenditures adjusted for price fluctuations based on cigarette prices from the Ministry of Internal Affairs and Communications' Retail Price Statistics Survey. Price increases occurred because of base price adjustments in April 2016 and April 2017, tobacco tax increases in October 2018, October 2020, and October 2021, and a revision of the consumption tax rate in 2019. The solid line shows fitted values from the interrupted time-series segmented regression model adjusted for seasonality and autocorrelation. The dashed line shows the predicted monthly tobacco expenditures per household if the COVID-19 pandemic and the first state of emergency had not occurred, assuming continuation of the pre-intervention trend (counterfactual). The time variable was centered at 0 for March 2020; thus, the intercept represents the estimated monthly expenditure in March 2020.

As a sensitivity analysis, we additionally fitted a model including a quadratic time term (time²). For monthly tobacco expenditures, the quadratic term was statistically significant (Table 3). In this model, the immediate level change after the first intervention was no longer statistically significant, whereas the post-first-intervention slope change became statistically significant.

The segmented regression model describes trends and changes in monthly tobacco expenditures across three phases: the period before the first state of emergency declaration, the period after the first declaration, and the period following the downgrading of COVID-19 to a common infectious disease.

Baseline trend: January 2015 to March 2020.

After the first state of emergency declaration: Level change — change from March 2020 to April 2020; Slope change — change in trend from before March 2020 to after April 2020.

After COVID-19 was downgraded to a common infectious disease: Level change — change from April 2023 to May 2023; Slope change — change in trend from April 2020 to April 2023 to after May 2023.

## Changes in smoking prevalences

The NDB Open Data included approximately 14–16 million men and 11–13 million women each year. The smoking prevalence among men exhibited a slight downward trend each year, declining from 34.2% in 2014 to 31.9% in 2022 (Fig 3). In contrast, the smoking prevalence among women remained relatively stable, fluctuating between 9.4% and 9.9% during the same period.

**Table 2.  Parameter estimates from the interrupted time-series segmented regression model for monthly tobacco expenditures adjusted for seasonality and autocorrelation.**

| | | Estimate | 95% confidence interval | | P-value |
| --- | --- | --- | --- | --- | --- |
| | | | Lower limit | Upper limit | |
| Intercept | | 797.883 | 724.578 | 871.187 | <0.001 |
| Baseline trend | | −2.501 | −4.037 | −0.964 | 0.002 |
| After the first state of emergency declaration | Level change | 132.004 | 42.281 | 221.727 | 0.005 |
| | Slope change | −1.294 | −5.039 | 2.451 | 0.486 |
| After COVID-19 was downgraded to a common infectious disease | Level change | −17.578 | −120.702 | 85.546 | 0.731 |
| | Slope change | 5.892 | 0.731 | 11.053 | 0.027 |
| February | | −51.703 | −115.314 | 11.908 | 0.110 |
| March | | 30.380 | −41.112 | 101.873 | 0.402 |
| April | | 3.547 | −69.484 | 76.577 | 0.923 |
| May | | 45.424 | −28.406 | 119.255 | 0.225 |
| June | | 46.667 | −27.042 | 120.376 | 0.212 |
| July | | 75.578 | 2.017 | 149.140 | 0.044 |
| August | | 60.454 | −12.967 | 133.875 | 0.106 |
| September | | 133.170 | 59.912 | 206.429 | <0.001 |
| October | | −45.150 | −118.046 | 27.747 | 0.222 |
| November | | −42.058 | −113.603 | 29.487 | 0.247 |
| December | | 9.502 | −55.873 | 74.877 | 0.774 |

Note: Seasonality was modeled using month indicators (February–December), with January as the reference category. Autocorrelation was accounted for using a linear mixed model with an AR(1) covariance structure. P-values were obtained from t-tests for fixed-effect parameters in the linear mixed model, using the denominator degrees-of-freedom method specified in SPSS. Values <0.05 were considered statistically significant.

## Discussion

### Cigarette sales volume

Data from national tobacco product sales in the United States indicate that cigarette sales increased by approximately 12% during the pandemic lockdown period, possibly reflecting increased consumption or stockpiling of tobacco products [23]. In Japan, there was no clear increase or decrease in cigarette sales before or after the first state of emergency was declared. However, the observed slowdown in the declining trend may suggest a sustained change in purchasing and use patterns after the onset of the pandemic, including stockpiling or changes in smoking intensity among continuing smokers, similar to patterns reported in the United States. This pattern could be explained by reduced shopping frequency and heightened uncertainty early in the pandemic that encouraged advance purchases, increased time spent at home that shifted smoking opportunities, and pandemic-related stress and anxiety that may have increased cigarette use among some smokers.

Consistent with a short-term shift in purchasing, price-adjusted monthly tobacco expenditures per household increased by 132 JPY in April 2020. Because this measure represents the average monthly expenditure across all households with two or more members, including households without current smokers, and reflects purchasing rather than direct consumption, the 132 JPY increase should be interpreted as modest in absolute terms, even though it corresponds to approximately a 17% increase relative to the pre-intervention level of about 800 JPY. This estimate should not be interpreted directly as the change per smoking household, because the denominator includes all households with two or more members, irrespective of smoking status. Using published Japanese household-based data indicating that 25.47% of households were classified as smoking households [24], the observed increase corresponds approximately to 518 JPY

**Table 3. Parameter estimates from the interrupted time-series segmented regression model for monthly tobacco expenditures including a quadratic time term, adjusted for seasonality and autocorrelation.**

| | | Estimate | 95% confidence interval | | P-value |
|---|---|---|---|---|---|
| | | | Lower limit | Upper limit | |
| Intercept | | 874.494 | 789.946 | 959.042 | <.001 |
| Baseline trend | | 5.140 | −0.128 | 10.408 | 0.055 |
| Quadratic time term (time²) | | 0.12 | 0.04 | 0.199 | 0.004 |
| After the first state of emergency declaration | Level change | 76.389 | −12.434 | 165.211 | 0.090 |
| | Slope change | −13.265 | −21.878 | −4.652 | 0.004 |
| After COVID-19 was downgraded to a common infectious disease | Level change | −27.931 | −120.688 | 64.827 | 0.545 |
| | Slope change | −2.513 | −9.759 | 4.734 | 0.485 |
| February | | −52.094 | −115.462 | 11.274 | 0.106 |
| March | | 30.795 | −38.214 | 99.805 | 0.379 |
| April | | 8.717 | −61.118 | 78.552 | 0.805 |
| May | | 51.888 | −18.503 | 122.279 | 0.147 |
| June | | 53.965 | −16.313 | 124.243 | 0.131 |
| July | | 83.462 | 13.291 | 153.634 | 0.020 |
| August | | 68.681 | −1.396 | 138.759 | 0.055 |
| September | | 141.494 | 71.507 | 211.480 | <.001 |
| October | | −37.003 | −106.837 | 32.831 | 0.296 |
| November | | −34.492 | −103.681 | 34.698 | 0.325 |
| December | | 15.440 | −49.707 | 80.587 | 0.639 |

Note: Seasonality was modeled using month indicators (February–December), with January as the reference category. Autocorrelation was accounted for using a linear mixed model with an AR(1) covariance structure. P-values were obtained from t-tests for fixed-effect parameters in the linear mixed model, using the denominator degrees-of-freedom method specified in SPSS. Values <0.05 were considered statistically significant.

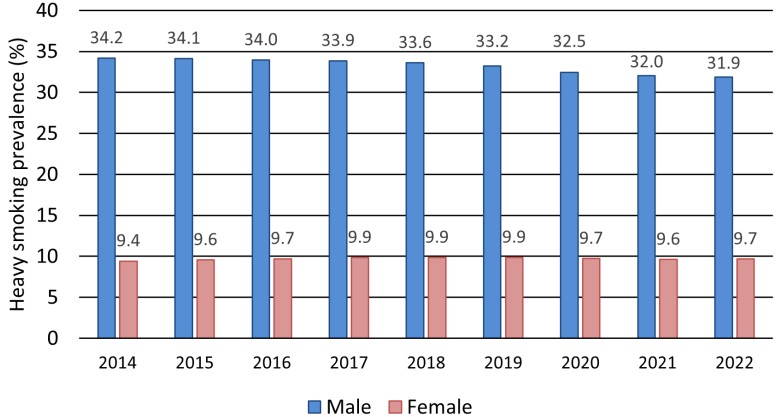

**Fig 3. Trends in the prevalence of heavy smoking among individuals aged 40 years and older.** Note: Heavy smokers are defined as individuals who have smoked more than 100 cigarettes in total or for more than 6 months and who reported smoking within the past month.

per smoking household, and the pre-intervention average corresponds approximately to 3,141 JPY per smoking household; however, these are only rough approximations and should be interpreted cautiously. These aggregate data cannot distinguish stockpiling from a true increase in cigarettes consumed per smoker. Moreover, because the Family Income

and Expenditure Survey excludes single-person households, the findings may have limited generalizability to the overall household population. According to the 2020 Population Census, single-person households accounted for 38.1% of general households in Japan [25], indicating that a substantial segment of household types was not represented in the expenditure analysis. Because tobacco purchasing and smoking patterns may differ in single-person households, the direction and magnitude of any resulting bias cannot be determined a priori.

The present study was limited to cigarette sales. In recent years, sales of heated tobacco products (HTPs) have increased, and their retail revenue now accounts for approximately 80% of that generated by conventional cigarettes [26]. Accordingly, substitution between cigarettes and HTPs may have affected the observed changes in cigarette sales and household tobacco expenditures and should be considered a potential confounder in the interpretation of the ITS results. One study conducted in Osaka, Japan, reported that during the state of emergency, only 2.2% of individuals switched from cigarettes to HTPs, while 0.6% switched in the opposite direction [10]. However, because this evidence was obtained in a single city during a limited period, it cannot be generalized to the national level over the longer study period.

## Possible changes in smoking behavior

In this ecological analysis, smoking intensity was not measured directly at the individual level; therefore, any changes in intensity can only be inferred indirectly from population-level indicators. Since the onset of the COVID-19 pandemic, the medical community and mass media have consistently warned the public that smoking increases the risk of severe illness and death. Therefore, it was anticipated that smoking intensity would decrease due to heightened public concern about the associated health risks. However, only a few studies have reported reductions in smoking intensity during this period [14,15 27]. In contrast, substantial evidence indicates that a greater number of individuals increased their smoking intensity rather than reduced it during the pandemic [28–39]. Similar trends were observed in Japan. Even among high-risk populations—such as individuals with heart disease—an increase in smoking intensity was reported among smokers [12,40]. In our study, the slowdown in the declining trend of cigarette sales and the immediate increase in the level of price-adjusted tobacco expenditures in April 2020 may reflect changes in tobacco purchasing patterns, such as stockpiling, and possibly changes in smoking intensity among continuing smokers; however, these data cannot distinguish between these explanations. Sensitivity analyses including a quadratic time term did not materially change the cigarette sales results, but the monthly tobacco expenditure results were less stable. In particular, for monthly tobacco expenditures, the immediate level change after the first intervention was no longer statistically significant when the quadratic term was included. Accordingly, the expenditure findings should be interpreted with greater caution than the cigarette sales findings. Moreover, after May 2023, the change in the expenditure trend was significant in the model adjusted for seasonality and autocorrelation, but this result was not robust in the quadratic sensitivity analysis, further supporting a cautious interpretation of the monthly tobacco expenditure series.

Mask-wearing may have played a role in shaping smoking opportunities and context, which could in turn affect smoking intensity among continuing smokers rather than smoking prevalence. In Japan, mask-wearing was strongly encouraged until March 2023, and the majority of individuals consistently wore masks beginning in April 2020, when the first state of emergency was declared [41]. Although there is limited evidence that mask use directly changed population-level cigarette sales, tobacco expenditures, or smoking prevalences, mask use may have reduced the convenience of smoking in public settings and contributed to shifts in where smoking occurred. Consistent with this possibility, reports of increased smoking at home—where mask use is not required—suggest that environmental constraints may have displaced smoking to private spaces, potentially influencing patterns of consumption without necessarily changing smoking prevalence [10].

## Implications for cessation and prevention

If individuals are unable to reduce or quit smoking despite the short-term risk of severe illness and mortality from COVID-19, then the long-term risk of developing malignancies 10 or 20 years after initiation may have limited influence on

behavior modification. In this sense, the pandemic provides a natural experiment: even under a heightened and immediate health threat, cessation may remain difficult at the population level. Many countries and regions have attempted to deter smoking by placing health warnings on cigarette packaging. The broader literature suggests that warnings alone—particularly those emphasizing long-term health risks and mortality—may have limited effectiveness in promoting cessation for some smokers. These observations underscore the limitations of risk communication alone and suggest that effective strategies should also address stress and coping mechanisms.

Furthermore, stress has been reported to influence smoking intensity [10 29 42,43]. Smoking control measures—such as displaying health warnings on cigarette packaging and restricting smoking areas—may inadvertently contribute to increased smoking intensity by inducing stress among smokers and the general population. Therefore, smoking cessation strategies that do not provoke stress may prove more effective. For instance, interventions that modify aspects of the choice architecture to influence behavior in a predictable manner—without restricting options or significantly altering economic incentives—such as "nudges [44]," are promising alternatives. Additionally, identifying the specific types of stress that trigger increased smoking may help develop more targeted prevention strategies.

Although a deceleration in the declining trend of tobacco expenditures was observed after May 8, 2023, when COVID-19 was reclassified as a common infectious disease and legal restrictions on individual behaviors were lifted, this change should be interpreted cautiously because it was not robust in the quadratic sensitivity analysis. The absence of robust evidence for a clear shift in tobacco expenditures may reflect Japan's prolonged economic stagnation [45] and a cultural tendency to avoid abrupt behavioral change.

### Smoking prevalence

In Japan, the smoking prevalence among men had been gradually declining even before the first state of emergency was declared. There was no substantial change in smoking prevalence following the declaration. Because the data represent nationwide trends, it remains unclear whether the lack of change indicates that the COVID-19 pandemic had no effect on smoking behavior, or whether increases in new smokers were offset by an approximately equal number of individuals who quit smoking. A survey conducted in Osaka, Japan, reported that approximately 12% of individuals quit smoking during the first state of emergency in April and May 2020 [10]. However, this study only assessed behavioral changes over a two-month period during which activity was heavily restricted; longer-term behavior remains unknown. Similarly, reports from other countries have suggested that lockdowns and the pandemic contributed to small reductions in smoking prevalence [16 27 30 35 46–49]. These studies showed that the reductions were modest in all cases, with only one report indicating a decrease greater than 5% [49]. Conversely, one study from the United States found that serious smoking cessation activity among adults declined sharply and remained suppressed for over a year during the pandemic [13]. Overall, behavioral restrictions associated with COVID-19 do not appear to have significantly reduced smoking. The suggestion that the number of new smokers equaled the number of individuals who quit following the declaration of the state of emergency in Japan is not strongly supported by the available evidence, suggesting that the pandemic's impact on national smoking prevalences was likely minimal.

Overall, the available evidence suggests that the COVID-19 pandemic and associated behavioral restrictions had, at most, a minimal impact on national smoking prevalence in Japan. However, because the NDB Open Data cover only adults aged 40 years and older, these findings should be interpreted cautiously and should not be generalized directly to younger adults aged 20–39 years. If younger adults were more responsive to pandemic-related social and environmental changes, the present data may have underestimated changes in smoking prevalence in the overall adult population.

### Limitations

This study has several limitations. First, we relied on aggregate indicators (cigarette sales, household tobacco expenditures, and national smoking prevalence), and thus the findings may be subject to ecological fallacy and should not

be interpreted as individual-level changes in smoking behavior. Second, sales and expenditure data reflect purchasing rather than actual consumption; therefore, the observed increase in April 2020 may partly reflect stockpiling or shifts in purchasing timing. In addition, the Family Income and Expenditure Survey covers only households with two or more members, and the expenditure estimates represent averages across all such households, including those without current smokers; accordingly, they should not be interpreted as direct estimates of change per smoking household. Published household-based data suggest that smoking households account for only a minority of Japanese households, and thus the implied increase per smoking household would be substantially larger than the all-household average, although such calculations are approximate. This may limit generalizability to single-person households, which accounted for 38.1% of general households in Japan in the 2020 Population Census and may have different tobacco purchasing and smoking patterns. Because single-person households comprised a large proportion of households, the expenditure findings should be interpreted primarily as applying to households with two or more members rather than to the entire household population. Third, our sales series did not include heated tobacco products or other nicotine products, and substitution across products may have confounded inferences based on cigarette sales and household tobacco expenditures. Because the market share of HTPs increased substantially during the study period, this limitation should be considered when interpreting the ITS results. Fourth, smoking prevalence was derived from NDB Open Data health checkups among individuals aged ≥40 years and may be affected by reporting bias and pandemic-related changes in participation. Accordingly, the prevalence findings may not be generalizable to the entire adult population and may underestimate changes if smoking behavior was more responsive among younger adults during the pandemic. Finally, as with interrupted time-series analyses in general, our interpretation assumes that no other major contemporaneous factors influenced the outcomes around the intervention points; residual confounding and model specification choices may have affected the estimates. Sensitivity analyses using a quadratic time term suggested that the monthly tobacco expenditure results were less robust than the cigarette sales results.

## Conclusion

During the first state of emergency in Japan, the declining trend in tobacco sales decelerated. Furthermore, household spending on tobacco increased by approximately 17% following the declaration, although this expenditure finding was less robust in the quadratic sensitivity analysis. In contrast, there was no significant change in smoking prevalence. From the early stages of the pandemic, smoking was believed to be closely associated with adverse outcomes of COVID-19. These health warnings could have prompted smokers to quit. However, they did not result in a substantial reduction in smoking prevalence and were more consistent with a shift in tobacco purchasing patterns, including stockpiling, and possible changes in smoking intensity among continuing smokers than with a meaningful population-level decline in smoking behavior. Moreover, there was no robust evidence of a clear reversal in these purchasing-related changes after May 2023 in the available data.

## Author contributions

**Conceptualization:** Kanae Kondo, Ichizo Morita.

**Data curation:** Kanae Kondo, Ichizo Morita.

**Formal analysis:** Kanae Kondo, Ichizo Morita.

**Methodology:** Kanae Kondo.

**Project administration:** Isao Ohsawa.

**Supervision:** Kanae Kondo, Ichizo Morita.

**Validation:** Shigemitsu Sakuma.

**Visualization:** Kanae Kondo, Ichizo Morita.

**Writing – original draft:** Kanae Kondo, Ichizo Morita.

**Writing – review & editing:** Shigemitsu Sakuma, Isao Ohsawa.

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
