## [Decision Letter · Decision Letter 0]

10 Apr 2026

PONE-D-26-02685Changes in tobacco sales before, during, and after the COVID-19 pandemic in Japan: an interrupted time series analysisPLOS One

Dear Dr. Kondo,

Thank you for submitting your manuscript to PLOS ONE. After careful consideration, we feel that it has merit but does not fully meet PLOS ONE’s publication criteria as it currently stands. Therefore, we invite you to submit a revised version of the manuscript that addresses the points raised during the review process.

We look forward to receiving your revised manuscript.

Kind regards,

Fabrizio Ferretti, Ph.D.

Academic Editor

PLOS One

Journal Requirements:

2. Please note that your Data Availability Statement is currently missing the repository name and the DOI/accession number of each dataset OR a direct link to access each database. If your manuscript is accepted for publication, you will be asked to provide these details on a very short timeline. We therefore suggest that you provide this information now, though we will not hold up the peer review process if you are unable.

Reviewers' comments:

Reviewer's Responses to Questions

**Comments to the Author**

1. Is the manuscript technically sound, and do the data support the conclusions?

Reviewer #1: Partly

Reviewer #2: Partly

2. Has the statistical analysis been performed appropriately and rigorously? 

Reviewer #1: Yes

Reviewer #2: No

3. Have the authors made all data underlying the findings in their manuscript fully available?

Reviewer #1: Yes

Reviewer #2: Yes

4. Is the manuscript presented in an intelligible fashion and written in standard English?

Reviewer #1: Yes

Reviewer #2: Yes

5. Review Comments to the Author

Reviewer #1: The authors used several data sources to provide evidence on changes in cigarette sales and smoking in Japan before and after the COVID-19 pandemic. The authors found increases in cigarette sales and purchase after the pandemic compared to the pre-pandemic levels and trends.

There are some major and minor concerns (not listed in order) that need to be considered.

1. Can you explain the data clearly, providing detailed information on what is available and how you used it in your analysis?

2. Model specification: I am not sure about the controls in the model, but the model should account for seasonality. For example, quarterly or monthly FE should be included in the models to account for seasonality. I suggest writing a formal model and defining terms.

3. Lines 104 to 106: Cite methodological studies that presented this approach as valid and how they are interpreted. Here is one of such studies: https://journals.sagepub.com/doi/abs/10.1177/00131640021970358

4. From Figure 1 and the results reported in Table 1, it is clear that the increase was not temporary. The effect of the pandemic on cigarette sales is a permanent change in the rate of decline, which lasted through 2024. I am not sure about how this was described as temporary. That change in slope never went away throughout the sample period. In Figure 2, we see a permanent shift in the level after the onset of the pandemic. That gap between the expected (counterfactual) and actual trends, although diminishing over time, persisted and even widened after May 2023.

5. Lines 205 – 207: Should be in the intro before setting up the hypothesis.

6. Line 217: Again, I don't know why the authors claim there was a temporary increase. You moved up to a different intercept and continued with the same trend. Levels never shifted to the original, so how is that temporary? It was a rather permanent increase in the level.

7. The figure title and captions in the text are more confusing and make reading difficult to distinguish. Add “Note:” to figure captions.

Reviewer #2: This paper uses segmented interrupted time series (ITS) models to analyze three national-level data sources (cigarette sales, household tobacco expenditures, and smoking prevalence) to examine how the COVID-19 pandemic affected smoking behavior in Japan. The authors report that the declining trend in cigarette sales decelerated after the first state of emergency (April 2020) and household tobacco expenditures spiked by approximately 19%, but smoking prevalence did not meaningfully change. The topic is policy-relevant, and the multi-data-source design is a strength. However, there are several concerns that should be addressed.

1. Autocorrelation and seasonality. Monthly and quarterly time-series data are expected to exhibit autocorrelation, and tobacco purchasing has well-documented seasonal patterns. Standard OLS-based segmented regression will produce incorrect standard errors and p-values if autocorrelation is not addressed. The manuscript mentions that only in the limitations section as a caveat, rather than having it tested or corrected. Seasonality was likewise not modeled. I recommend the authors account for autocorrelation and seasonality adjustments, and provide standard diagnostic tests (e.g., Durbin-Watson statistic, Ljung-Box test) for the residuals.

2. Absence of sensitivity analyses. There is no exploration of alternative model specifications, for example, different polynomial functions (now is all assumed linear). ITS results can be sensitive to polynomial choices.

3. Overstatement of conclusions relative to the data. The abstract and conclusion state that "smoking behavior among smokers appeared to intensify" and that these effects "persisted beyond May 2023." Neither claim is adequately supported by the data results. First, the sales/expenditure variables measure purchasing, not consumption. The authors themselves acknowledge that "these data cannot distinguish stockpiling from a true increase in cigarettes consumed per smoker," yet the abstract and conclusion reassert the intensification interpretation. Second, the "persistence" claim rests on a non-significant result at the second intervention point. This is a type II error concern: the non-significant estimate reflects absence of evidence, not evidence of persistence. I recommend the authors revise the abstract and conclusions to focus on changes in sales/purchasing patterns (even just to be more consistent with your title), with the “intensifying” and “persistence” interpretations discussed just as possibilities rather than conclusive findings.

4. The 153 JPY increase is translated to a ~19% rise, but the pre-treatment average of 800 JPY is calculated across all households with two or more members, even including non-smoking households. To aid interpretation, the authors should report (or estimate from published data) what fraction of Japanese households include a current smoker. This would allow readers to gauge the approximate per-smoker-household effect. Additionally, since it also excludes single-person households, the authors should note what proportion of Japanese households are single-person (e.g., approximately 38% as of the 2020 Population Census of Japan) and discuss how this exclusion may affect generalizability. Particularly, single-person households may have very different smoking patterns.

5. The authors note that heated tobacco products (HTPs) are excluded from the analysis, though its revenue now represents approximately 80% of conventional cigarette revenue in Japan, a substantial market shift occurring over the study period. If smokers transitioned between cigarettes and HTPs during the pandemic, the observed changes in the cigarette sales/expenditure could be partially attributable to cross-product substitution. I know that the authors cite one Osaka survey showing low switching rates (2.2%), but this is limited evidence from a single city during a short window and cannot be generalized to the national level over a longer time period. This limitation should be discussed more prominently and explicitly acknowledged as a potential confounder in the ITS results, rather than treated as a secondary limitation.

6. The NDB Open Data include only adults aged 40+, excluding younger adults (i.e., 20–39), a demographic likely with very different tobacco use patterns and pandemic-related behavioral changes compared to older adults. In Japan, smoking prevalence among younger men has historically been higher than among older age groups, and younger adults may be more responsive to social and environmental changes. This limitation is mentioned only briefly; it should be more explicitly flagged when interpreting the prevalence trends, and the authors should discuss the direction in which this exclusion might bias their findings.

6. PLOS authors have the option to publish the peer review history of their article (what does this mean?). If published, this will include your full peer review and any attached files.

Reviewer #1: No

Reviewer #2: No

---

## [Author Response · Author response to Decision Letter 1]

18 May 2026

Reviewer #1: The authors used several data sources to provide evidence on changes in cigarette sales and smoking in Japan before and after the COVID-19 pandemic. The authors found increases in cigarette sales and purchase after the pandemic compared to the pre-pandemic levels and trends.

There are some major and minor concerns (not listed in order) that need to be considered.

Dear Reviewer 1,

Thank you very much for your careful review and thoughtful comments on our manuscript. We sincerely appreciate the time and effort you devoted to evaluating our work. Your comments were extremely helpful and led us to improve the manuscript substantially.

In the process of revision, additional analyses became necessary in response to the reviewers’ comments. During this period, newer data also became available. Accordingly, we updated the data sets and re-conducted the analyses using the most recent data available.

Specifically, for the number of cigarette sales, the study period was updated from the first quarter of 2015 to the first quarter of 2025 to the first quarter of 2015 to the fourth quarter of 2025. For National tobacco expenditures, the study period was updated from January 2015 to May 2025 to January 2015 to February 2026.

These updates are reflected in the revised manuscript and in our point-by-point response. We are very grateful for your valuable comments, which have helped us strengthen the manuscript.

1. Can you explain the data clearly, providing detailed information on what is available and how you used it in your analysis?

Thank you for this helpful comment. The Data sources section was revised to clarify what information was available from each dataset and how it was used in the present study. Specifically, the revised text now states that the cigarette sales analysis used the quarterly number of cigarettes sold, that the household expenditure analysis used price-adjusted monthly tobacco expenditures for households with two or more members, and that smoking prevalence data were annual percentages and therefore were summarized descriptively rather than analyzed in the interrupted time-series model.

The following sentences were added to the Data sources section of the revised manuscript:

New (P3, L95) In this study, the quarterly number of cigarettes sold was used for the analysis of cigarette sales trends.

New (P4L, 106) Thus, the monthly analysis was based on price-adjusted tobacco expenditures for households with two or more members.

New (P4L, 113) Because these data were available as annual national percentages, smoking prevalence in men and women was summarized descriptively rather than analyzed using the interrupted time-series model.

2. Model specification: I am not sure about the controls in the model, but the model should account for seasonality. For example, quarterly or monthly FE should be included in the models to account for seasonality. I suggest writing a formal model and defining terms.

Thank you for this helpful comment. We agree that autocorrelation and seasonality should be handled explicitly in the present time-series analyses. In response, we revised the analytic approach to account for seasonality using quarter indicators for the quarterly sales data and month indicators for the monthly expenditure data, and to account for autocorrelation using linear mixed models with an AR(1) covariance structure.

The following sentences were added.

New (P4, L133): For quarterly cigarette sales, seasonality was modeled using quarter indicators (Q2–Q4), with Q1 as the reference category. For monthly tobacco expenditures, seasonality was modeled using month indicators (February–December), with January as the reference category. To account for autocorrelation, the models were fitted using linear mixed models with an AR(1) covariance structure.

The results tables have been revised as follows.

Table 1. Parameter estimates from the interrupted time-series segmented regression model for quarterly cigarette sales adjusted for seasonality and autocorrelation.

95% confidence interval

Estimate Lower limit Upper limit P-value

Intercept 244.982 228.278 261.685 <0.001

Baseline trend -9.991 -11.214 -8.767 <0.001

After the first state of emergency declaration Level change -17.468 -41.080 6.143 0.133

Slope change 7.962 4.858 11.067 <0.001

After COVID-19 was downgraded to a common infectious disease Level change -1.481 -29.119 26.157 0.910

Slope change -0.924 -5.227 3.379 0.643

Q2 22.254 8.958 35.551 0.002

Q3 42.881 29.759 56.002 <0.001

Q4 19.110 5.973 32.248 0.006

Note: Seasonality was modeled using quarter indicators (Q2–Q4), with Q1 as the reference category. Autocorrelation was accounted for using a linear mixed model with an AR(1) covariance structure. P-values were obtained from t-tests for fixed-effect parameters in the linear mixed model, using the denominator degrees-of-freedom method specified in SPSS. Values <0.05 were considered statistically significant.

Table 2. Parameter estimates from the interrupted time-series segmented regression model for monthly tobacco expenditures adjusted for seasonality and autocorrelation.

95% confidence interval

Estimate Lower limit Upper limit P-value

Intercept 797.883 724.578 871.187 <0.001

Baseline trend -2.501 -4.037 -0.964 0.002

After the first state of emergency declaration Level change 132.004 42.281 221.727 0.005

Slope change -1.294 -5.039 2.451 0.486

After COVID-19 was downgraded to a common infectious disease Level change -17.578 -120.702 85.546 0.731

Slope change 5.892 0.731 11.053 0.027

February -51.703 -115.314 11.908 0.110

March 30.380 -41.112 101.873 0.402

April 3.547 -69.484 76.577 0.923

May 45.424 -28.406 119.255 0.225

June 46.667 -27.042 120.376 0.212

July 75.578 2.017 149.140 0.044

August 60.454 -12.967 133.875 0.106

September 133.170 59.912 206.429 <0.001

October -45.150 -118.046 27.747 0.222

November -42.058 -113.603 29.487 0.247

December 9.502 -55.873 74.877 0.774

Note: Seasonality was modeled using month indicators (February–December), with January as the reference category. Autocorrelation was accounted for using a linear mixed model with an AR(1) covariance structure. P-values were obtained from t-tests for fixed-effect parameters in the linear mixed model, using the denominator degrees-of-freedom method specified in SPSS. Values <0.05 were considered statistically significant.

The figures presenting the results have been revised as follows.

New: Fig 1. Trend in cigarette sales before and after the first state of emergency declaration.

Note: The solid line shows fitted values from the interrupted time-series segmented regression model adjusted for seasonality and autocorrelation. The dashed line shows the predicted number of cigarette sales had the COVID-19 pandemic and the first state of emergency not occurred, assuming continuation of the pre-intervention trend (counterfactual). The time variable was centered at 0 for January–March 2020; thus, the intercept represents the estimated cigarette sales in that quarter.

New: Fig 2. Trend in monthly tobacco expenditures per household (households with two or more members) before and after the first state of emergency declaration.

Note: The round markers indicate tobacco expenditures adjusted for price fluctuations based on cigarette prices from the Ministry of Internal Affairs and Communications’ Retail Price Statistics Survey. Price increases occurred because of base price adjustments in April 2016 and April 2017, tobacco tax increases in October 2018, October 2020, and October 2021, and a revision of the consumption tax rate in 2019. The solid line shows fitted values from the interrupted time-series segmented regression model adjusted for seasonality and autocorrelation. The dashed line shows the predicted monthly tobacco expenditures per household if the COVID-19 pandemic and the first state of emergency had not occurred, assuming continuation of the pre-intervention trend (counterfactual). The time variable was centered at 0 for March 2020; thus, the intercept represents the estimated monthly expenditure in March 2020.

The relevant text in the Results section has been revised as follows.

Old (P5, L118): Using a segmented regression ITS model with two intervention points (April–June 2020 and April–June 2023), the level change at the first intervention was not statistically significant, whereas the slope became significantly less negative after the first declaration (p = 0.002), indicating a deceleration in the declining trend. In the first quarter of 2020, cigarette sales showed little change compared to the previous quarter; however, the post-intervention trend corresponded to a smaller quarterly decline (approximately 300 million cigarettes) compared with the pre-intervention period.

Nwe (P5, L166): After adjustment for seasonality and autocorrelation, the level change at the first intervention was not statistically significant, whereas the slope became significantly less negative after the first declaration (p < 0.001), indicating a deceleration in the declining trend. In the first quarter of 2020, cigarette sales showed little change compared to the previous quarter; however, the post-intervention trend corresponded to a smaller quarterly decline (approximately 200 million cigarettes) compared with the pre-intervention period.

Nwe (P5, L174): The quarter indicators were statistically significant, supporting the presence of seasonal variation in cigarette sales.

Old (P6, L142): From January 2015 to March 2020, price-adjusted monthly tobacco expenditures exhibited a declining trend with periodic fluctuations, decreasing from approximately 1,000 JPY to 800 JPY (Fig 2, Table 2). In April 2020, when the government declared the first state of emergency, price-adjusted monthly tobacco expenditures rose sharply by 153 JPY compared to the previous month (p < 0.001). However, from that point through April 2023, the expenditures continued to decline; the change in slope after April 2020 was not statistically significant (p = 0.251), indicating no evidence that the rate of decline differed from the pre-pandemic trend. After May 2023, when COVID-19 was downgraded to a common infectious disease, the rate of decline in price-adjusted monthly tobacco expenditures slowed, although the change was not statistically significant.

Nwe (P6, L208): From January 2015 to March 2020, price-adjusted monthly tobacco expenditures exhibited a declining trend with periodic fluctuations, decreasing from approximately 1,000 JPY to 800 JPY (Fig 2, Table 2). After adjustment for seasonality and autocorrelation, in April 2020, when the government declared the first state of emergency, price-adjusted monthly tobacco expenditures increased significantly by 132 JPY compared to the previous month (p = 0.005). However, from that point through April 2023, the expenditures continued to decline; the change in slope after April 2020 was not statistically significant (p = 0.486), indicating no evidence that the rate of decline differed from the pre-pandemic trend. After May 2023, when COVID-19 was downgraded to a common infectious disease, the rate of decline in price-adjusted monthly tobacco expenditures became significantly less steep (p = 0.027), whereas the level change was not statistically significant (p = 0.731).

Several month indicators were also statistically significant, supporting the presence of seasonal variation in tobacco expenditures.

3. Lines 104 to 106: Cite methodological studies that presented this approach as valid and how they are interpreted. Here is one of such studies: https://journals.sagepub.com/doi/abs/10.1177/00131640021970358

Thank you for this helpful comment. We revised the Statistical methods section to clarify the interpretation of the segmented regression coefficients. Specifically, we now state explicitly that the level-change parameter represents the immediate change at the first post-intervention time point relative to the value expected from the trend in the immediately preceding period, whereas the slope-change parameter represents the change in trend after the intervention relative to that preceding period. We also added the methodological reference by Huitema and McKean to support this clarification.

The following sentences and reference were changed and added at the Statistical methods.

New (P5, L149)

Following previous methodological guidance on segmented regression design specification [22], the level-change parameter for each intervention was defined as the immediate deviation at the first post-intervention time point from the value expected under continuation of the trend in the immediately preceding period, whereas the slope-change parameter was defined as the change in trend after the intervention relative to that immediately preceding period.

22. Huitema BE, McKean JW. Design specification issues in time-series intervention models. Educ Psychol Meas. 2000;60(1):38-58. https://doi.org/10.1177/0013164002197035

4. From Figure 1 and the results reported in Table 1, it is clear that the increase was not temporary. The effect of the pandemic on cigarette sales is a permanent change in the rate of decline, which lasted through 2024. I am not sure about how this was described as temporary. That change in slope never went away throughout the sample period. In Figure 2, we see a permanent shift in the level after the onset of the pandemic. That gap between the expected (counterfactual) and actual trends, although diminishing over time, persisted and even widened after May 2023.

Thank you for this important comment. The term “temporary” was imprecise and has been revised. Specifically, the description of cigarette sales was modified to avoid implying a temporary increase and to better reflect the sustained deceleration in the declining trend after the onset of the pandemic. In addition, the description of household tobacco expenditures was revised from “temporary increase” to “immediate increase in the level” in April 2020, which is more consistent with Figure 2 and Table 2.

The following sentences were changed to the Discussion section.

Old (P7, L192): However, the observed slowdown in the declining trend may suggest changes in purchasing and use patterns, including stockpiling or a temporary increase in smoking intensity among continuing smokers, similar to patterns reported in the United States.

New (P9, L306): However, the observed slowdown in the declining trend may suggest a sustained change in purchasing and use patterns after the onset of the pandemic, including stockpiling or changes in smoking intensity among continuing smokers, similar to patterns reported in the United States.

Old (P8, L227): In our study, the slowdown in the declining trend of cigarette sales and the temporary increase in price-adjusted tobacco expenditures immediately after April 2020 may reflect changes in purchasing and use patterns, such as stockpiling and/or a short-term increase in smoking intensity among continuing smokers; however, these data cannot distinguish between these explanations.

New (P10, L364): In our study, the slowdown in the declining trend of cigarette sales and the immediate increase in the level of price-adjusted tobacco expenditures in April 2020 may reflect changes in tobacco purchasing patterns, such as stockpiling, and possibly changes in smoking intensity among continuing smokers; however, these data cannot distinguish between these explanations.

Old (P10, L309): However, they did not result in a substantial reduction in smoking prevalence and were more consistent with a shift in purchasing and smoking patterns among continuing smokers—such as temporary stockpiling and/or a modest increase in smoking intensity—than with a meaningful population-level decline in smoking behavior.

New (P13, L537): However, they did not result in a substantial reduction in smoking prevalence and were more consistent with a shift in tobacco purchasing patterns, including stockpiling, and possible changes in smoking intensity among continuing smokers than with a meaningful population-level decline in smoking behavior.

5. Lines 205 – 207: Should be in the intro before setting up the hypothesis.

Th

---

## [Editor Report · Decision Letter 1]

19 May 2026

Changes in tobacco sales before, during, and after the COVID-19 pandemic in Japan: an interrupted time series analysis

PONE-D-26-02685R1

Dear Dr. Kondo,

We’re pleased to inform you that your manuscript has been judged scientifically suitable for publication and will be formally accepted for publication once it meets all outstanding technical requirements.

Kind regards,

Fabrizio Ferretti, Ph.D.

Academic Editor

PLOS One
---

## [Editor Report · Acceptance letter]

PONE-D-26-02685R1

PLOS One

Dear Dr. Kondo,

I'm pleased to inform you that your manuscript has been deemed suitable for publication in PLOS One. Congratulations! Your manuscript is now being handed over to our production team.

Kind regards,

on behalf of

Dr. Fabrizio Ferretti

Academic Editor

PLOS One